

# *Diegoaelurus*, a new machaeroidine (Oxyaenidae) from the Santiago Formation (late Uintan) of southern California and the relationships of Machaeroidinae, the oldest group of sabertooth mammals

Shawn P. Zack[1,2], Ashley W. Poust[3,4] and Hugh Wagner[3]

[1] Department of Basic Medical Sciences, University of Arizona College of Medicine—Phoenix, Phoenix, AZ, United States of America
[2] School of Human Evolution and Social Change, Arizona State University, Tempe, AZ, United States of America
[3] Department of Paleontology, San Diego Natural History Museum, San Diego, CA, United States of America
[4] University of California Museum of Paleontology, University of California, Berkeley, Berkeley, CA, United States of America

Corresponding authors
Shawn P. Zack,
zack@email.arizona.edu
Ashley W. Poust, apoust@sdnhm.org

## ABSTRACT

Machaeroidinae is a taxonomically small clade of early and middle Eocene carnivorous mammals that includes the earliest known saber-toothed mammalian carnivores. Machaeroidine diversity is low, with only a handful of species described from North America and Asia. Here we report a new genus and species of machaeroidine, *Diegoaelurus vanvalkenburghae*, established on the basis of a nearly complete dentary with most of the dentition from the late Uintan (middle Eocene) portion of the Santiago Formation of southern California. The new taxon is the youngest known machaeroidine and provides the first evidence for the presence of multiple machaeroidine lineages, as it differs substantially from *Apataelurus kayi*, the only near-contemporaneous member of the group. Phylogenetic analysis indicates that *Diegoaelurus* is the sister taxon of *Apataelurus*, while older species are recovered as a monophyletic *Machaeroides*. Both phylogenetic results are relatively weakly supported. The new taxon extends the record of machaeroidines to the end of the Uintan, potentially tying machaeroidine extinction to the faunal turnover spanning the middle to late Eocene transition in North America.

## INTRODUCTION

Machaeroidine oxyaenids are a small radiation of carnivorous eutherian mammals known from the early and middle Eocene of North America and Asia (*Matthew, 1909*; *Scott, 1937*; *Gazin, 1946*; *Dawson et al., 1986*; *Zack, 2019a*; *Zack, 2019b*). Machaeroidines have the distinction of being the oldest known saber-toothed mammalian carnivores. Postcranial morphology is most consistent with an arboreal or scansorial locomotor repertoire

**Table 1** Published specimens of Machaeroidinae, arranged by age and geography.

| Specimen | Taxon | Location | Formation | Age | Reference |
|---|---|---|---|---|---|
| CM 45115 | *Machaeroides simpsoni* (T) | Wyoming | Wind River | Wa7 | *Dawson et al. (1986)* |
| CM 36397 | *Machaeroides simpsoni* | Wyoming | Wind River | Wa7 | *Dawson et al. (1986)* |
| CM 37342 | *Machaeroides simpsoni* | Wyoming | Wind River | Br1 | *Dawson et al. (1986)* |
| AMNH FM 12644[a] | *Machaeroides eothen* (T) | Wyoming | Bridger | Br2 | *Matthew (1909)* |
| AMNH FM 11523 | *Machaeroides eothen* | Wyoming | Bridger | Br2 | *Matthew (1909)* |
| USNM 17059 | *Machaeroides eothen* | Wyoming | Bridger | Br2[b] | *Gazin (1946)* |
| USNM 361372 | *Machaeroides eothen* | Wyoming | Bridger | Br2 | *Tomiya et al. (2021)* |
| AMNH FM 12083 | *Machaeroides* cf. *M. eothen* | Wyoming | Bridger | Br3 | *Denison (1938)* |
| USNM 173514 | Machaeroidinae indet. | Wyoming | Bridger | Br3 | *Zack (2019a)* |
| FMNH PM 1506 | Machaeroidinae indet. | Wyoming | Washakie | Ui1 | *Tomiya et al. (2021)* |
| CM 11920 | *Apataelurus kayi* (T) | Utah | Uinta | Ui2 | *Scott (1937)* |
| SDSNH 38343 | *Diegoaelurus vanvalkenburghae* (T) | California | Santiago | Ui3 | This study |
| CM 2386 | Machaeroidinae indet. | Utah | Uinta | Ui3 | *Zack (2019a)* |
| ZIN 32759 | *Isphanatherium ferganensis* (T) | Kyrgyzstan | Lower Alai | ?Ir | *Lavrov & Averianov (1998)* |
| IVPP V7997 | *Apataelurus pishigouensis* (T) | Henan, China | Hetaoyuan | Ir | *Tong & Lei (1986)* |

Notes.
[a] Mislabeled as AMNH FM 11523 by *Matthew, (1909*, fig. 71).
[b] *Gazin (1946)* stated that USNM 17059 was from the base of Bridger C (Br3), but subsequently (*Gazin, 1976*) indicated that both USNM specimens of *M. eothen* are from the lower Bridger Formation.
Abbreviations: Br, Bridgerian; Ir, Irdinmanhan; T, type specimen; Ui, Uintan; Wa, Wasatchian.

(*Gazin, 1946*; *Zack, 2019a*), suggesting that these specialized carnivores preferred closed habitats. The record of machaeroidines is extremely scanty. Depending on the possible machaeroidine status of *Isphanatherium ferganensis* from the early or middle Eocene of Kyrgyzstan (*Lavrov & Averianov, 1998*; *Zack, 2019b*), there are either four or five named machaeroidine species, each represented by no more than a handful of specimens, along with a handful of additional specimens in open nomenclature (Table 1).

Asian machaeroidines are limited to the middle Eocene *Apataelurus pishigouensis* from the Hetaoyuan Formation of Henan Province, China (*Tong & Lei, 1986*; *Zack, 2019b*) and the tentatively referred *I. ferganensis*, each represented by a single specimen. The oldest known North American machaeroidine is early Eocene *Machaeroides simpsoni*, documented from the latest Wasatchian (Wa7) and earliest Bridgerian (Br1) of the Wind River Formation, Wind River Basin, Wyoming (*Dawson et al., 1986*). A second species of *Machaeroides*, *M. eothen*, is present in the middle Bridgerian (Br2, late early and earliest middle Eocene) of the Bridger Formation, Bridger Basin, Wyoming (*Matthew, 1909*; *Gazin, 1946*). The late Bridgerian (Br3, middle Eocene) machaeroidine record is limited to a dentary briefly mentioned by *Denison* (*1938*, p. 181) as an advanced variant of *M. eothen* and a postcranial skeletal association (USNM 173514, see *Zack, 2019a*), both also from the Bridger Formation. *Matthew* (*1909*, p. 462) also mentions an isolated m1 from this level, but the identity of this specimen is uncertain. The early Uintan (Ui1) record is similarly scanty, comprising a partial lower molar from the Washakie Formation, Washakie Basin, Wyoming (FMNH PM 1506, see *Tomiya et al., 2021*). None of the latter specimens is complete enough to permit identification to genus, although FMNH PM 1506, combining

a fully trenchant talonid with retention of a small metaconid, appears to be intermediate in morphology between *Machaeroides eothen* and the more derived *Apataelurus kayi* (*Tomiya et al., 2021*). The middle Uintan (Ui2) machaeroidine record comprises *Apataelurus kayi*, named for a pair of associated dentaries from the Wagonhound Member of the Uinta Formation, Uinta Basin, Utah (*Scott, 1937*; *Scott, 1938*). *Zack (2019a)* described a skeletal association from the late Uintan (Ui3) Myton Member of the Uinta Formation. *Wagner (1999)* briefly reported on the specimen described here, from the late Uintan of California. There is at least one additional record of a machaeroidine from the Uinta Basin (*Rasmussen et al., 1999*), but it is not clear if it derives from middle or late Uintan levels.

We report here on a new machaeroidine from the late Uintan (Ui3) portion (Member "C") of the Santiago Formation, San Diego County. The new taxon is represented by a nearly complete dentary preserving much of the dentition, which documents a smaller taxon than *Apataelurus kayi*, previously the only named Uintan machaeroidine. The specimen is the first record of a North American machaeroidine from the west coast or from any area outside of Utah and Wyoming (Fig. 1). The new taxon extends our knowledge of machaeroidine diversity and permits an exploration of the phylogenetic interrelationships of this rare group of early mammalian carnivores.

## MATERIALS & METHODS

Dental terminology follows *Rana et al. (2015)* with modifications from *Zack (2019b)*. All measurements were taken to the nearest hundredth of a millimeter using Mitutoyo digital calipers but are reported to the nearest tenth of a millimeter to avoid the impression of spurious accuracy.

To produce the 3D imaging a Nikon D750 camera with a Tokina Macro 100 mm lens was used to take 380 photos from different angles *via* a camera tethered to a computer *via* Smart Shooter 3. Models were then generated with Agisoft Metashape Standard Edition. The correct size and orientation of the 3D models was adjusted with Meshmixer and the OBJ file was converted to a CTM file and a PLY file with Meshlab. The CTM file is available for viewing and download on the San Diego Natural History Museum website (https://3dfiles.sdnhm.org/api/?specimen=38343&name=38343_Dentary_RT&extension=ctm). The CTM file can also be accessed *via* the Museum's collections catalog (https://www.sdnhm.org/science/paleontology/resources/collection-database/). The PLY file can be accessed on Morphosource (https://www.morphosource.org/projects/000415654), Project 415654.

**Phylogenetic Methods**.—To date, the only rigorous phylogenetic analysis to include machaeroidines (*Zack, 2019a*) focused on the placement of Machaeroidinae within the larger radiation of early Cenozoic carnivorous eutherians, recovering machaeroidines as oxyaenids. No study has investigated machaeroidine interrelationships. Unfortunately, the matrix used by *Zack (2019a)* is not appropriate for this task, as most features that vary within Machaeroidinae are not included in the matrix. This primarily reflects the fact that many features that vary within Machaeroidinae are not parsimony informative with regard to the sample used by *Zack (2019a)*. To investigate relationships within

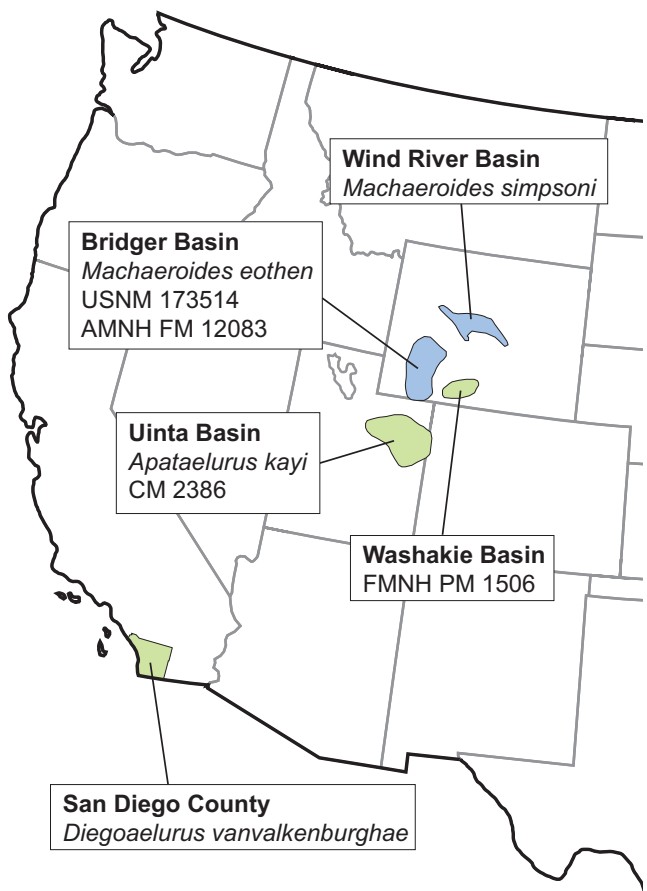

**Figure 1 Distribution of North American representatives of Machaeroidinae.** Light blue and green indicate basins producing early and middle Eocene machaeroidines, respectively. Note that while the Bridger Basin has produced several specimens of early Eocene *Machaeroides eothen*, USNM 173514 and AMNH FM 12083 are earliest middle Eocene. Basin outlines adapted from *Dickinson et al. (1988)* and *Tomiya et al. (2021)*.

Machaeroidinae, a new character taxon matrix was constructed. The matrix samples five machaeroidine taxa (*Machaeroides eothen*, *M. simpsoni*, *Apataelurus kayi*, *A. pishigouensis*, and the new taxon described below) and two outgroups (the oxyaenid *Dipsalidictis krausei* and the hyaenodontid *Prototomus phobos*) scored for 24 dental and mandibular characters. Among named machaeroidines, *Isphanatherium ferganensis* was excluded because known material, an isolated M1, has limited overlap with other taxa, as the locus is otherwise only documented in the two species of *Machaeroides*. North American machaeroidines described in open nomenclature were also excluded as this material is fragmentary and/or has limited overlap with better-known taxa.

All characters are binary except for character 8, which is treated as unordered. Only eleven characters are parsimony informative for the current sample but additional characters are included to facilitate future investigations as machaeroidine diversity becomes better known. The character list, specimen list, and matrix are accessible on MorphoBank as

project P4091 (https://morphobank.org/index.php/Projects/ProjectOverview/project_id/4091). The character list, specimen list, and matrix are also available in the Supplemental Information.

The matrix was analyzed in TnT version 1.5 (*Goloboff & Catalano, 2016*). Due to the small number of taxa under consideration, an exhaustive search was performed using the TnT's implicit enumeration algorithm. Ingroup monophyly was enforced *via* inclusion of a dummy character of arbitrarily high weight that was deactivated prior to calculation of tree statistics.

The electronic version of this article in Portable Document Format (PDF) will represent a published work according to the International Commission on Zoological Nomenclature (ICZN), and hence the new names contained in the electronic version are effectively published under that Code from the electronic edition alone. This published work and the nomenclatural acts it contains have been registered in ZooBank, the online registration system for the ICZN. The ZooBank LSIDs (Life Science Identifiers) can be resolved and the associated information viewed through any standard web browser by appending the LSID to the prefix http://zoobank.org/. The LSID for this publication is: urn:lsid:zoobank.org:pub:9946B7AE-5EB5-45BF-953E-B8C245AC806C. The online version of this work is archived and available from the following digital repositories: PeerJ, PubMed Central and CLOCKSS.

## RESULTS

### Systematic paleontology

MAMMALIA *Linnaeus, 1758*
EUTHERIA *Huxley, 1880*
OXYAENODONTA *Van Valen, 1971*
OXYAENIDAE *Cope, 1877*
MACHAEROIDINAE *Matthew, 1909*
*DIEGOAELURUS* gen. nov.
urn:lsid:zoobank.org:act:2BE5B4DE-1808-44C4-BF56-89090012ACE6

**Type Species**.—*Diegoaelurus vanvalkenburghae* sp. nov.
**Diagnosis**.—Autapomorphies of *Diegoaelurus* are loss of p1; p2 single, rather than double-rooted; presence of a paraconid on p3; and a relatively deep mandibular flange.
**Differential Diagnosis**.—Differs from *Machaeroides* in lacking metaconids on m1-2 and in having lower, more elongate trigonids on m1-2. Differs from *Apataelurus* in having the mandibular flange only extend back to the level of p2, a relatively larger p3, m2 only moderately larger than m1, and coronoid process taller than m2.
**Etymology**.—*Diego*, for San Diego, California and Gr. -*aelurus*, cat. While -*ailurus* is more frequently used as a combining form, -*aelurus* is used to align with the closely related *Apataelurus*.

**Comments.**—In the phylogenetic analysis presented below, *Diegoaelurus* is recovered as the sister taxon of *Apataelurus*, raising the question of why *D. vanvalkenburghae* is placed in a new genus rather than *Apataelurus*. There are substantial differences in the morphology of the limited overlapping material of *D. vanvalkenburghae* and species of *Apataelurus*. Some of the contrasts, particularly the more expansive mandibular flange in *Apataelurus*, suggest substantial differences in the ecology of the two genera. Additionally, the synapomorphies uniting *D. vanvalkenburghae* with *Apataelurus* describe loss of lower molar metaconids and entocristids, both features that have independently evolved numerous times in the evolution of hypercarnivorous mammals. Given the strong homoplastic tendencies in these traits, support for monophyly of *Diegoaelurus* plus *Apataelurus* should be considered weak, requiring confirmation from additional material of one or both taxa.

*DIEGOAELURUS VANVALKENBURGHAE* sp. nov.
urn:lsid:zoobank.org:act:91F6F357-5E5F-4D5B-BBBD-FE66AB3DF8ED
(Figs. 2–5, Table 2)

**Holotype.**—SDSNH 38343, right dentary preserving i2, c, p3, and m1-2.

**Type Locality.**—SDSNH locality 3276 (Jeff's Discovery), Oceanside, San Diego County, California. As described by *Walsh (1996)*, the Jeff's Discovery assemblage comprises several adjacent localities first discovered by Mr. Jeff Dahlgren. The Jeff's Discovery fauna has produced a distinctive fauna including a number of taxa otherwise rare in the San Diego Eocene. Aside from *Diegoaelurus vanvalkenburghae*, distinctive elements of the Jeff's Discovery fauna include the hyaenodontid *Limnocyon* sp., carnivoran *Tapocyon dawsonae*, tapiroid *Hesperaletes borineyi*, and basal artiodactyl *Ibarus* sp. (*Walsh, 1996*; *Wesley & Flynn, 2003*; *Colbert, 2006*).

**Stratigraphy and Age.**—Member "C" of the Santiago Formation, late Uintan (Ui3) North American Land Mammal Age (NALMA), middle Eocene (*Walsh, 1996*).

**Diagnosis.**—As for the genus.

**Etymology.**—Named in honor of Dr. Blaire Van Valkenburgh, in recognition of her substantial contributions to our understanding of iterative evolution in carnivorous mammals and saber-tooth paleoecology.

**Description.**—The dentary is intact along most of the horizontal ramus, but is broken posteriorly, with portions of the ascending ramus, including the condylar process and posterior margin of the coronoid process missing (Fig. 2).

A flange projects ventrally and somewhat laterally (Figs. 2A, 3), extending the depth of the anterior dentary by almost a third. The shape of the flange is curved, contrasting with the triangular flange in *Machaeroides*, and its posterior edge is slightly concave marking it as a very distinct feature and contrasting with a linear posterior margin in *Machaeroides*. A thin ridge borders it anteriorly, narrowing to a thin raised blade along the anterior surface below the canine. No mental foramina are present immediately adjacent to the symphysis on the anterior aspect of the jaw, unlike in *Machaeroides simpsoni*, though three are present on the lateral surface (Fig. 2A), one directly below the p2 alveolus, one placed

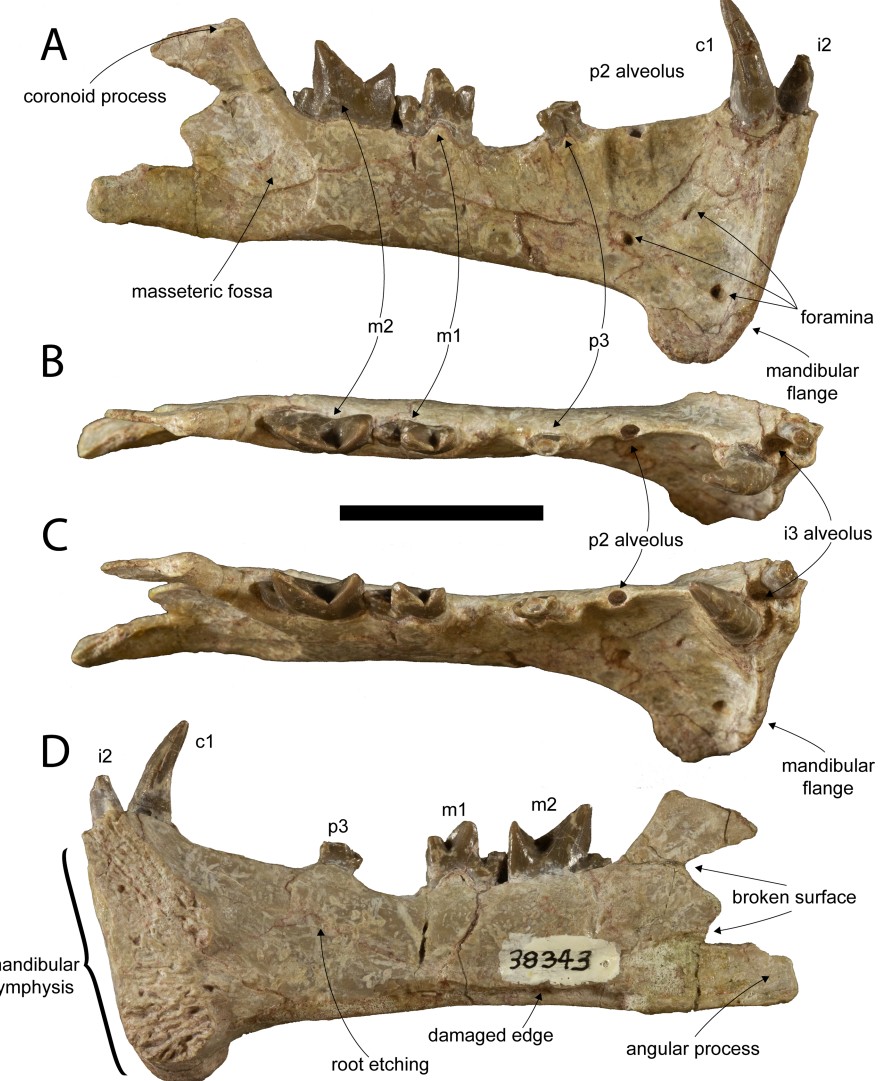

**Figure 2** Holotype of *Diegoaelurus vanvalkenburghae* sp. nov. (SDSNH 38343). Right dentary with i2, c, p3, m1-2, and alveoli for i3 and p2 in (A) buccal, (B) occlusal, (C) oblique occlusobuccal, and (D) lingual views. Scale bar is 10 mm.

anteroventrally just above the flange, and a third, smaller foramen about mid-height at the level of the distal margin of the canine. The two larger foramina are directed laterally, while the third is directed anterodorsally, at a very shallow angle to the dentary surface. The placement of these foramina differs from *Machaeroides eothen* where one foramen is more anterior, very close to the front of the jaw. This difference may relate to the expansion of the flange itself in SDNHS 38343.

The symphysis is well developed and vertically oriented, with a straight anterior margin and a bilobate posterior (Fig. 2D). The lineations on the symphyseal surface trend in two different directions. In the lower lobe along the ventral portion of the flange and opposite

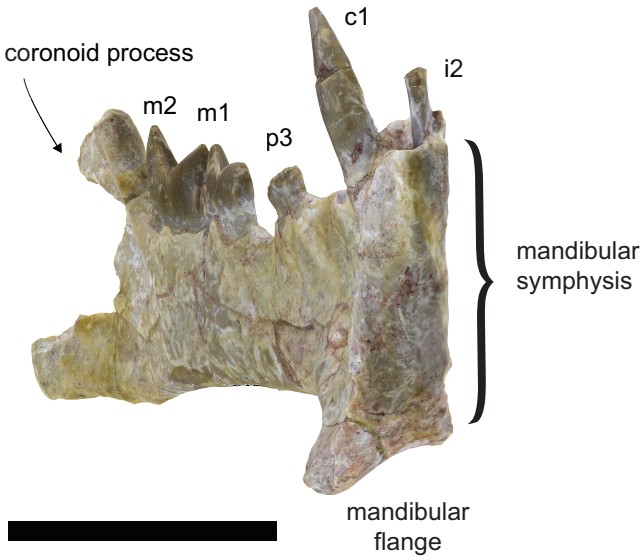

**Figure 3** **Anterior view of the holotype of *Diegoaelurus vanvalkenburghae* sp. nov. (SDSNH 38343) showing the lateral flaring of the mandibular flange.** Image is taken from a 3D model of the holotype. Scale bar is 10 mm.

the upper portion of the flange, the lineations run strongly anterior/posteriorly, with even the lowest few angled down anteriorly below the plane of the horizontal. The upper lobe by contrast has linear features that splay outward and upward, primarily at about a 60 degree angle from horizontal spanning from the midline to the alveoli of the incisors. The plane of the symphysis is only slightly angled away from the long axis of the dentary, meaning the jaw lacks the wide anterior expansion seen in *M. simpsoni*.

The horizontal ramus posterior to the flange is rectangular and transversely compressed. Between the p3 and the m1 the dorsal edge of the mandible dips down, a feature which may represent the effects of long finished bone resorption and filling of the alveolus and possibly support the idea that the adult p4 tooth was lost during early life. A furrow visible on the medioventral edge represents a break that exposed the interior cortical bone (Fig. 2D). Interestingly, this broken edge is filled with sediment; the break was not sustained during final erosion or collection, rather during biostratinomy, or perhaps a prior episode of erosion and redeposition. Given the rather complete nature of the specimen, presence of teeth, and lack of surface weathering this is an oddly contradictory taphonomic feature. Though slightly damaged the angular process is present giving us the length of the jaw, but unfortunately the condyle and much of the coronoid process are broken (Figs. 2A, 2D). The anterior and ventral borders of the masseteric fossa are clearly defined, showing that it would have been a deeply incised and large area of muscle attachment in life. The anterior margin of the masseteric fossa extends to a level below the back of the trigonid of m2. The dorsal margin of the fossa is less incised than the other borders and ends only slightly below the preserved portion of the coronoid process. Although the posterior margin of the coronoid process is absent, it appears to represent nearly the full height, extending to a

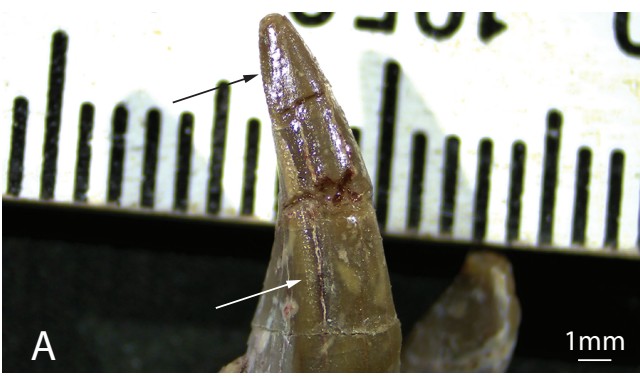

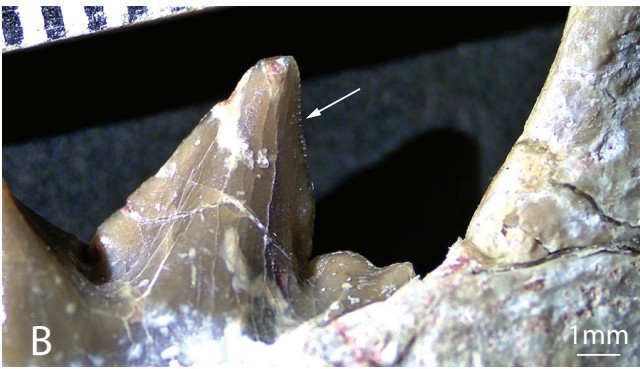

**Figure 4** **Serrations on the teeth of *Diegoaelurus vanvalkenburghae* sp. nov. (SDSNH 38343).** (A) Left canine in buccal view showing serrations on the distal carina of the lower canine (arrows); (B) left m2 in lingual view showing serrations on the distal carina of the protoconid (arrow). Scale bars are 1 mm.

level just above the height of the m2. Such a low coronoid process is typical of saber-tooth carnivores (*Matthew, 1910*; *Emerson & Radinsky, 1980*).

As in other machaeroidines, the dentary preserves evidence of only two lower incisors, the more mesial represented by a root and partial crown and the more distal by an alveolus (Figs. 2B–2C, 3). There is a small space mesial to the first incisor, but there is no evidence of an alveolus for a third incisor. Reduction in the number of lower incisors is shared with other machaeroidines in which this region is preserved: *Apataelurus kayi* and *Machaeroides eothen* (*Scott, 1938*; *Gazin, 1946*). *Denison (1938)* hypothesized that i1 was lost in *A. kayi*, and, in the absence of contradictory evidence, the lower incisors of *D. vanvalkenburghae* are tentatively considered i2-3.

Much of the apical crown of i2 is broken (Figs. 2–3). The preserved portion of the tooth, including the root, is compressed mesiodistally and slightly inclined buccally. The crown appears to have been relatively tall. The buccal face of the crown is gently convex and vertical, while the lingual face is flat and slopes basolingually. The lingual surface is wider than the buccal surface, making the crown wedge-shaped in occlusal view. The preserved mesial and distal margins are vertical and parallel, but not enough of the crown is preserved

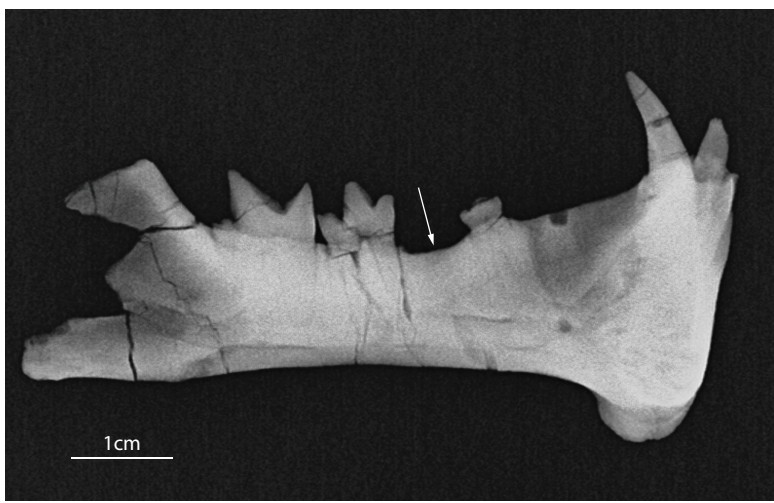

**Figure 5  Radiograph of the left dentary of *Diegoaelurus vanvalkenburghae* sp. nov. (SDSNH 38343).**
Note the absence of alveoli in the expected position of p4 (arrow) indicating that the tooth was either pathologically absent or that loss occurred well antemortem. Scale bar is 1 cm.

**Table 2  Measurements (mm) of the holotype of *Diegoaelurus vanvalkenburghae*.**

|  | SDSNH 38343 |
| --- | --- |
| Length of dentary | 71.5 |
| Length of dentition (i2 alveolus-m2) | 51.4 |
| Length of cheek dentition (p2 alveolus-m2) | 35.2 |
| Depth at flange | 23.9 |
| Depth of flange below line of jaw | 5.3 |
| Depth below m1 | 11.6 |
| m2 length | 10.4 |
| width | 3.8 |
| height | 8.0 |
| m1 length | 7.9 |
| width | 3.0 |
| height | 6.7 |
| p3 length | 4.5 |
| width | 2.1 |
| p2 length (alveolus) | 2.5 |
| width (alveolus) | 1.7 |
| c1 length | 4.7 |
| width | 3.5 |
| height | 13.4 |
| i2 length | 3.3 |
| width | 1.8 |
| height | 5.1 |

to determine if it was pointed or more spatulate. There is no development of basal cuspids or cingulids, and the preserved structure of the crown is simple.

The alveolus of i3 is immediately distal to i2 (Figs. 2–3). The alveolus indicates that the tooth was somewhat larger than i2, while the shape of the alveolus indicates that i3 was also compressed mesiodistally.

The unworn lower canine is nearly completely preserved, a first for Machaeroidinae (Figs. 2–3). The tooth is substantially larger than either incisor and would also have been much taller than i2 if the latter were complete. The canine is implanted vertically, lacking the buccal inclination of i2. As with the incisors, the crown of c1 is mesiodistally compressed, with an oblique buccolingual axis, oriented posteromedially relative to the dentary. The crown is tall and recurved lingually, tapering to a sharp point. The buccal surface is convex, while the lingual surface is formed by mesiolingual and distolingual faces that meet at a distinct angle, forming a peak that runs down the center of the lingual surface of the crown. Where the buccal and lingual surfaces of the crown meet, blunt, mesial and distal carinae are present. Serrations extend the length of the distal carina (Fig. 4A), while the mesial carina lacks serrations. The distal carina is stronger and extends further basally than the mesial carina. There is no development of cingulids or a basal heel, but the thin enamel extends modestly further basally on the buccal and lingual surfaces of the crown than on the mesial or distal surfaces.

A large diastema separates the canine from p3 (Figs. 2A–2C). Within this diastema, near the midpoint but closer to p3 than to the canine, is a single, small alveolus, for a single-rooted p2. The size of the alveolus indicates that p2 was likely the smallest lower tooth, approximately half the size of i2.

The first preserved cheek tooth is a double-rooted p3 (Fig. 2). The protoconid is broken near its base, but the remainder of the crown is intact. The tooth is approximately 57% the length of m1, relatively larger than in *Apataelurus kayi* (42%) but substantially reduced relative to *Machaeroides eothen* (88%). Mesially, the crown has a small but differentiated paraconid, deflected slightly lingually from the protoconid. This contrasts with both *M. eothen* and *A. kayi*, which lack paraconids on p3. The protoconid was large, dominating the crown, although its height cannot be determined. In cross-section, the broken base of the protoconid is lenticular. Distal to the protoconid, the talonid bears a single cusp, a large, low hypoconid positioned close to the buccal margin of the crown and aligned with the distal carina of the protoconid. The hypoconid has a steep buccal and more shallow lingual slope. Lingual to the hypoconid, the margin of the talonid is slightly concave at the distolingual corner of the crown, with very faint cuspids evident on either side of the concavity.

Immediately posterior to p3, p4 is absent. Alveoli for p4 are not visible externally nor are they evident in a radiograph of the specimen (Fig. 5), indicating that the absence of p4 reflects either pathology or *in vivo* loss well prior to death rather than postmortem damage. The size of the gap between p3 and m1 indicates that p4 was substantially longer than p3, but shorter than m1 and much shorter than m2.

Posterior to the missing p4, m1 is an elongate, narrow tooth (Fig. 2). The trigonid of m1 is bicuspid, with no trace of a metaconid. The paraconid is transversely compressed and

relatively elongate and low. The paraconid portion of the paracristid is oriented obliquely mesiolingually, with the apex of the paraconid mesiolingual to the protoconid apex. The crest slopes up from the carnassial notch, but the lingual half is flat topped due to wear. On the buccal surface of the paraconid, close to its mesial margin, there is a short, vertical mesiobuccal cingulid, defining an embrasure for the back of the p4 talonid. The cingulid is restricted to this vertical portion and does not continue around the base of the cusp. On the lingual side of the trigonid, the paraconid is separated from the protoconid by a deep groove extending lingually and slightly distally from the carnassial notch.

The protoconid of m1 is larger and substantially taller than the paraconid and is also transversely flattened. Two crests descend its apex. The trenchant protoconid portion of the paracristid is oriented mesially and slightly buccally and descends from the apex to the carnassial notch. The more weakly defined protocristid is directed distally and descends the protoconid vertically towards the talonid.

The talonid of m1 is much shorter than the trigonid and lower than both trigonid cusps. Buccally, there is no hypoflexid separating the trigonid from the talonid. The talonid is narrow and trenchant, with no entoconid or entocristid lingually. The talonid is dominated by the hypoconid, which has a vertical buccal slope and a somewhat gentler lingual slope. The hypoconid is basically flat-topped, but a low, centrally positioned apex is present. Running directly mesial from the hypoconid apex is a short cristid obliqua. This crest ends relatively abruptly at the mesial end of the talonid, where it is separated from the protocristid by a small carnassial notch. Nearly directly distal to the hypoconid, and connected by a weak postcristid, is a small hypoconulid. The hypoconulid is slightly lower than the hypoconid and centrally positioned on the distal margin of the crown. Aside from the mesiobuccal cingulid, there is no development of cingulids on m1.

The m2 of SDSNH 38343 is larger than m1 (Fig. 2). The trigonid of m2 is equivalent in length to the entirety of m1, and the trigonid cusps on m2 are considerably taller than those on m1. Aside from size, the morphology of the trigonid of m2 is very similar to m1, lacking a metaconid and with a nearly longitudinal paracristid. Additionally, the protoconid of m2 is more distally reclined than that of m1, with a vertical distal carina and more elongate mesial crest, resulting in a relatively longer paracristid than on m1. The distal carina bears serrations, similar in morphology to the canine (Fig. 4B). As with m1, there is a small, vertical mesiobuccal cingulid that helps define an embrasure for the m1 hypoconulid.

The talonid of m2 is relatively smaller and lower than on m1. The basic structure of the talonid is similar, with a hypoconid and hypoconulid, and no entoconid or entocristid. However, the hypoconulid is more lingually positioned relative to the hypoconid, with the result that the postcristid is oblique rather than longitudinal. The hypoconulid is also slightly larger than on m1, and its apex is better separated from the hypoconid.

**Comparisons.**—The type specimen of *Diegoaelurus vanvalkenburghae* can be directly compared with all named machaeroidines except *Isphanatherium ferganensis*, which is known only from an isolated upper molar. Numerous features of the lower dentition and dentary link *D. vanvalkenburghae* to other machaeroidines, including a broadened symphyseal region of the dentary with a ventral mandibular flange below the canine and mesial premolars, low coronoid process, reduced premolars mesial to p4, loss of m3,

and m1-2 with hypercarnivorous features including open trigonids, elongate, sectorial paracristids, and reduced metaconids and talonids.

Compared with the earliest known machaeroidine, *Machaeroides simpsoni*, *D. vanvalkenburghae* has a deeper mandibular flange with a more rounded ventral margin, but the anteroposterior length of the flange is similar in the two taxa. The posterior dentary is unknown in *M. simpsoni*, preventing comparisons, as is the lower canine. *Machaeroides simpsoni* retains a small, double-rooted p1, which is lost in *D. vanvalkenburghae*, and a double-rooted p2, which is single-rooted in the San Diego taxon. Known lower molars of *M. simpsoni* are fragmentary, but the relative molar sizes in the two species do appear to be similar, though m2 does appear to be proportionally larger in *D. vanvalkenburghae*. Both taxa have elongate, longitudinal paracristids, consistent with hypercarnivory. On the trigonids, *M. simpsoni* retains small but distinct metaconids, which are lacking in *D. vanvalkenburghae*. In general, the morphology of *Machaeroides simpsoni* appears less hypercarnivorously adapted than that of *Diegoaelurus vanvalkenburghae*.

The younger type species of *Machaeroides*, *M. eothen*, is better known than *M. simpsoni*. As with the latter species, *M. eothen* has a less developed, more triangular mandibular flange that is similar in length to *D. vanvalkenburghae*. More posteriorly, the coronoid process is somewhat taller than in *D. vanvalkenburghae*. The lower canine is incompletely preserved in *M. eothen*, inhibiting comparisons to *D. vanvalkenburghae*. Serrations are absent from the distal carina of the canine of USNM 17059, but this surface is worn from contact with the upper canine (*Gazin, 1946*, p. 343). Again resembling *M. simpsoni*, *Machaeroides eothen* has a double-rooted p1 and p2, differing from *D. vanvalkenburghae*, which lacks p1 and has a single rooted p2. The crown of p3 is preserved in *M. eothen*. It differs from *D. vanvalkenburghae* in lacking a paraconid and in being larger relative to the remainder of the dentition. As with *M. simpsoni*, m1-2 of *M. eothen* are more similar in size than in *D. vanvalkenburghae*, in which m2 is distinctly larger than m1. The trigonids of *M. simpsoni* are taller and less elongate than in the San Diego form. Again as in *M. simpsoni*, *M. eothen* retains metaconids on m1-2, unlike *D. vanvalkenburghae*. Talonid morphology is documented in the Bridgerian taxon, which retains low entocristids, contrasting with *D. vanvalkenburghae*, which has fully trenchant talonids. Talonid size is similar. As was the case with *M. simpsoni*, the morphology of *Machaeroides eothen* is similar to but less hypercarnivorously adapted than that of *Diegoaelurus vanvalkenburghae*.

Comparisons with the Chinese machaeroidine *Apataelurus pishigouensis* are limited due to the fragmentary nature of the type and only specimen of this taxon, a dentary fragment with p4-m1. Most of the dentary is not preserved, and the shape of the mandibular flange cannot be evaluated. However, the flange appears to have been larger than in *D. vanvalkenburghae*, extending to a point beneath the mesial margin of p4. The m1s of both species are similar in having a relatively low, elongate trigonid. Both taxa also lack metaconids. Unlike *D. vanvalkenburghae*, the talonid of *A. pishigouensis* is very short, comprising approximately a quarter of the length of the tooth. Based on the comparisons that can be made, both *D. vanvalkenburghae* and *A. pishigouensis* are relatively hypercarnivorous machaeroidines. However, the apparent difference in the size

 

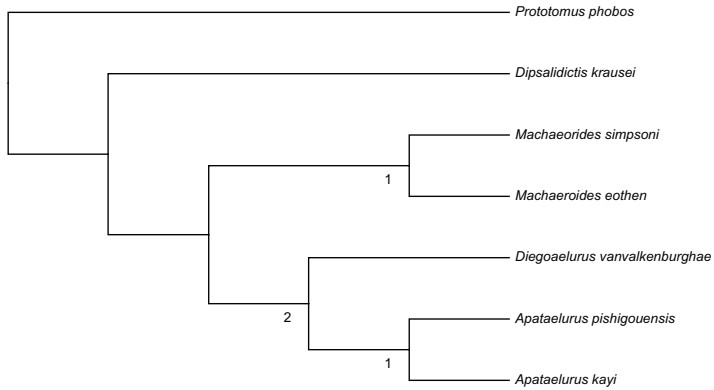

**Figure 6  Phylogeny of Machaeroidinae.** Single most parsimonious tree (L: 30; CI: 0.83; RI: 0.61) depicting the interrelationships of Machaeroidinae. Numbers below nodes within the ingroup indicate Bremer support.

of the dentary flange suggests differences in saber morphologies and, potentially, in feeding ecology.

The type species of *Apataelurus*, *A. kayi*, is better known than *A. pishigouensis*, permitting more extensive comparisons with *Diegoaelurus vanvalkenburghae*. The mandibular flange of *A. kayi* is incompletely preserved, but it extends to the level of p4, indicating a larger flange than in *D. vanvalkenburghae*. The coronoid process of *A. kayi* is much lower than in *D. vanvalkenburghae*, lower than the trigonid of m2. A single-rooted p1 is retained in *A. kayi*, contrasting with the loss of this tooth in *D. vanvalkenburghae*. Similarly, *A. kayi*, has a double-rooted p2, contrasting with the single-rooted tooth of *D. vanvalkenburghae*. The premolars of *A. kayi* are closely spaced and separated from the canine by a large diastema. In contrast, p2 is separated from p3 by a distinct diastema in *D. vanvalkenburghae*. As a result, there is less of a postcanine diastema in the San Diego form, despite the loss of p1, and it is possible shortening of this portion of the dentary helps to explain the loss of p1 in *D. vanvalkenburghae*. More posteriorly, *A. kayi* and *D. vanvalkenburghae* share a reduced p3, although *A. kayi* lacks the paraconid present in *D. vanvalkenburghae*. The m2 of *Apataelurus kayi* is much larger relative to m1 than in *D. vanvalkenburghae*. Both taxa share low, open trigonids on both molars, and both lack metaconids and entocristids. However, *A. kayi* has much more reduced talonids, with a short m1 talonid and a rudimentary, unicuspid m2 talonid, contrasting with the large, bicuspid talonid present in *D. vanvalkenburghae*. Overall, while *Diegoaelurus vanvalkenburghae* is advanced relative to *Apataelurus kayi* in its degree of premolar reduction, the remainder of the morphology of the Utah taxon appears more specialized. As with *A. pishigouensis*, the apparently larger mandibular flange of *A. kayi* suggests further differences in saber morphology that are presently undocumented.

**Phylogenetic Results.**—Analysis of the character-taxon matrix produced a single most parsimonious tree (L: 30; CI: 0.83; RI: 0.61) (Fig. 6). The analysis recovers a monophyletic *Machaeroides* as the sister taxon to a clade comprising *Diegoaelurus* and a monophyletic *Apataelurus*. Of the three ingroup nodes, only the node subtending middle Eocene

machaeroidines (*Apataelurus* plus *Diegoaelurus*) has Bremer support greater than one, with two additional steps required to recover trees that do not include this grouping.

Monophyly of *Machaeroides* is supported by a single unambiguous synapomorphy, presence of a double-rooted p1 (character 8, state 1). In contrast, *Apataelurus kayi* and both outgroups have a single-rooted p1 (state 1), while *D. vanvalkenburghae* lacks p1 (state 2). Monophyly of middle Eocene machaeroidines is supported by two unambiguous synapomorphies, loss of the metaconid on m1-2 (character 17, state 1) and absence of an entocristid on m1 (character 18, state 1). Monophyly of *Apataelurus* is supported by two unambiguous synapomorphies, reduction of the length of the m1 talonid to approximately $\frac{1}{4}$ the length of the crown (character 19, state 1) and posterior expansion of the mandibular flange to a point below p3 (character 21, state 1).

Unambiguous autapomorphies are only identified in two of the five ingroup taxa. *Machaeroides eothen* has seven autapomorphies. However, all but one of these is a feature of the upper dentition, which cannot be scored for *Diegoaelurus* or either species of *Apataelurus*. The remaining feature reconstructed as an autapomorphy of *M. eothen* is the presence of a symmetrical p4 protoconid, which contrasts with *M. simpsoni*, both *Apataelurus* species, and the oxyaenine outgroup. Three autapomorphies are reconstructed for *D. vanvalkenburghae*. Two of these describe reduction of the anterior premolars: loss of p1 (character 8, state 2) and presence of a single-rooted p2 (character 9, state 1). The third autapomorphy of *D. vanvalkenburghae* is the presence of a paraconid on p3 (character 11, state 1). Neither species of *Apataelurus* is reconstructed as having any autapomorphies. This is initially surprising given the distinctive morphology of *A. kayi*. However, the fragmentary nature of *A. pishigouensis* renders optimization of most of these features ambiguous. Thus, features such as an enlarged m2 (character 16, state 1), reduced m2 talonid (character 17, state 1), and low coronoid process (character 23, state 1) are equally parsimonious as *Apataelurus* synapomorphies or *A. kayi* autapomorphies. Presence of a tall p4 paraconid (character 13, state 1) and broad p4 talonid (character 14, state 1) is shared by both *Apataelurus* species but cannot be evaluated in *Diegoaelurus*, while a mandibular condyle positioned below the toothrow (character 24, state 1) cannot be evaluated in either *D. vanvalkenburghae* or *A. pishigouensis*. One distinctive feature of *Diegoaelurus*, presence of a rounded border to the mandibular flange (character 22, state 1) cannot be evaluated in either *Apataelurus* species.

## DISCUSSION

**Machaeroidine Phylogeny and Diversity.**—Discussion of any aspect of machaeroidine biology and evolution is hampered by the small number of machaeroidine specimens known and their generally fragmentary nature. This is particularly true of any attempt to identify patterns in machaeroidine evolution. Most named species are represented by single specimens, and even the best-represented taxon, *Machaeroides eothen*, is known from only four or five individuals (Table 1), making assessment of individual variation difficult. Moreover, the anatomy of most named taxa is poorly known. The upper dentition is documented in only two species also represented by the lower dentition (*Gazin, 1946*;

*Dawson et al., 1986*), and cranial and postcranial anatomy is documented in only one taxon for which the dentition is preserved (*Gazin, 1946*). With these caveats, preliminary comments can be made concerning the phylogenetic interrelationships and diversity of machaeroidines based on the results of the phylogenetic analysis.

The results of the phylogenetic analysis are partially concordant with stratigraphy (Fig. 7). Middle Eocene machaeroidines (Uintan and Irdinmanhan) are monophyletic with respect to early and earliest middle Eocene taxa (Wasatchian and Bridgerian). However, recovery of a monophyletic *Machaeroides* implies a ghost lineage of approximately nine million years for the clade comprising *Diegoaelurus* and *Apataelurus*. Monophyly of *Machaeroides* is supported by a single feature, presence of a double-rooted p1, and could easily be overturned by future discoveries. In particular, the morphology of the upper dentition of *M. eothen* is substantially derived relative to *M. simpsoni*, while the upper dentition is unknown in species of *Apataelurus* and *Diegoaelurus*. Additionally, the specimen described as an ''advanced variant'' of *M. eothen* from Br3 (AMNH FM 12083) has partially fused p1 roots (*Denison, 1938*), potentially documenting a transition to the reduced p1 morphologies present in *Apataelurus* and *Diegoaelurus*.

If the upper dentition of middle Eocene machaeroidines proves to be similarly derived, it would likely overturn the result recovered here. One line of evidence that this may be the case comes from the unnamed Uintan machaeroidine described by *Zack (2019a)*. The edentulous maxillary fragment of that specimen (CM 2386) has a very reduced lingual M1 root, suggestive of a reduced protocone. This would correspond with state 1 of character 6, one of the features that distinguishes *M. eothen* from *M. simpsoni*.

Both *Diegoaelurus vanvalkenburghae* and *Apataelurus kayi* have multiple autapomorphies, providing the first clear evidence of parallel machaeroidine lineages. Distinctive features of *D. vanvalkenburghae* concern its reduction of the anterior premolars, with p1 lost and p2 single-rooted and strongly reduced. Autapomorphic features of *A. kayi* are more numerous, including expansion of the mandibular flange back to the level of the distal premolars (shared with *A. pishigouensis*), enlargement of m2 relative to m1, reduction of the talonid of m2, and lowering of the coronoid process to below the apex of m2. Although *D. vanvalkenburghae* or its ancestor is not yet recorded in Ui2, it is highly likely that two machaeroidine lineages were present during this interval. Given the very limited fossil record of Machaeroidinae, it is quite likely that our understanding of machaeroidine standing diversity remains incomplete.

Aside from *D. vanvalkenburghae* and *Apataelurus kayi*, additional machaeroidine records have been described from Uintan-aged deposits of western North America (Fig. 7: *Rasmussen et al., 1999*; *Zack, 2019a*; *Tomiya et al., 2021*). The possibility that some of these could represent *D. vanvalkenburghae* should be briefly considered. The most straightforward of these to evaluate is FMNH PM 1506, a partial right ?m1 from the Twka2 interval of the Washakie Formation (Ui1b), Washakie Basin, Wyoming (*Tomiya et al., 2021*). FMNH PM 1506 is similar to *D. vanvalkenburghae* in size and in talonid morphology, but it differs in retaining a small metaconid. For this reason, it likely represents a distinct taxon, although nothing in the little that is known of its morphology would preclude a relationship to *D. vanvalkenburghae*. A partial machaeroidine skeleton,

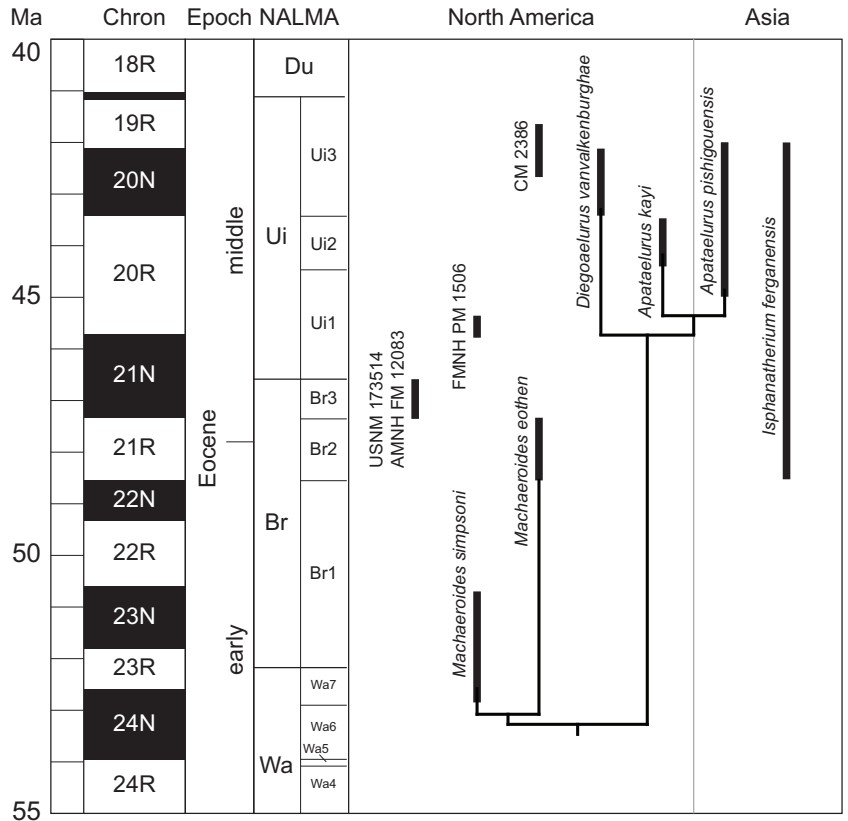

**Figure 7** **Temporally calibrated machaeroidine phylogeny.** Topology of sampled taxa as in Fig. 5 Temporal positions of additional machaeroidines, including *Isphanatherium ferganensis* and taxa in open nomenclature, are shown. Biochronological assignments follow *Gazin (1976)*, *Dawson et al. (1986)*, *Walsh (1996)*, *Averianov & Godinot (2005)*, *Gunnell et al. (2009)*, *Zack (2019a)*, *Zack (2019b)*, and *Tomiya et al. (2021)*. Temporal assignment of biochrons follows *Tsukui & Clyde (2012)*, *Chew & Oheim (2013)*, *Murphey et al. (2018)*, *Wang et al. (2019)*, and *Tomiya et al. (2021)*.

CM 2386, recently described from the Myton Member of the Uinta Formation (Ui3), Uinta Basin, Utah (*Zack, 2019a*), represents a small machaeroidine, much smaller than *Apataelurus kayi* from the same depositional basin and, on the basis of size, could plausibly represent *D. vanvalkenburghae*. Unfortunately, there is no anatomical overlap between CM 2386 and SDSNH 38343, making comparisons impossible. In view of the distinctiveness of California Uintan faunas in comparison with those from the Western Interior (*Lillegraven, 1980*; *Walsh, 1996*; *Walsh, 2000*; *Atwater & Kirk, 2018*), it would be premature to assign CM 2386 to *D. vanvalkenburghae* in the absence of comparable material. Finally, *Rasmussen et al. (1999)* mentioned but did not describe or provide a specimen number for an additional machaeroidine from the Uinta Basin, characterized only as distinct from *A. kayi*. Based on such limited information, it is impossible to determine whether this material represents *D. vanvalkenburghae*.

The phylogenetic analysis also provides support for *Zack*'s (*2019b*) assignment of the former *Propterodon pishigouensis* to *Apataelurus*. The Chinese taxon shares a posteriorly

expanded mandibular flange and reduced m1 talonid with *Apataelurus kayi*, in addition to sharing loss of the metaconid on m1 with both *A. kayi* and *D. vanvalkenburghae*. By nesting *A. pishigouensis* within Machaeroidinae, the topology recovered by the analysis supports a North American origin for the species, as all other sampled taxa are from North America. However, consideration of the other named Asian machaeroidine, *Isphanatherium ferganensis*, suggests the potential for more complex scenarios of machaeroidine biogeography and evolution.

The type and only known specimen of *I. ferganensis* is an isolated upper molar, interpreted as M2 by *Lavrov & Averianov (1998)*, but more likely representing M1 (*Zack, 2019b*). Only three characters used in the phylogenetic analysis can be scored for *I. ferganensis*, and these cannot be scored for most other members of the ingroup and for these reasons, *I. ferganensis* was not included in our analysis. Based on *Lavrov & Averianov (1998)* descriptions and illustrations, *I. ferganensis* displays a mixture of primitive and derived features in comparison with species of *Machaeroides*, the other taxa in which M1 is preserved. The protocone is strongly reduced (character 6, state 1), comparable to *M. eothen* and greater than *M. simpsoni*, while the metastyle is more elongate and longitudinal (character 5, state 1) than in either species of *Machaeroides*. Alongside these apparently derived features, the paracone and metacone appear to be less fused than in either *M. eothen* or *M. simpsoni* (character 4, state 0).

If the morphology of *I. ferganensis* has been accurately assessed, it suggests parallel hypercarnivorous developments in Asia and North America. Specifically, the presence of a paracone and metacone that are less fused in *I. ferganensis* than either species of *Machaeroides* could indicate that the derived hypercarnivorous features of M1 shared by *I. ferganensis* and *M. eothen* evolved in parallel from an ancestor with an upper molar morphology more primitive than in any known machaeroidine.

The broader implications of the morphology of *Isphanatherium* are dependent on its relationship to machaeroidines known only from lower dentitions. One intriguing possibility is that *I. ferganensis* is related to *A. pishigouensis*, the only other known Asian machaeroidine. The two taxa are similar in size and both demonstrate a more hypercarnivorous morphology than either species of *Machaeroides*. The two species are also likely to be similar in age (Fig. 7). Shipigou, the type locality of *A. pishigouensis*, is correlated to the Irdinmanhan ALMA (middle Eocene) (*Wang et al., 2019*). The age of Andarak-2, which produced *I. ferganensis*, is less certain, but the mammalian fauna is most supportive of an Irdinmanhan assignment (*Averianov & Godinot, 2005*). If *Isphanatherium* and *Apataelurus* are closely related, this would raise the possibility of dispersal of *Apataelurus* from Asia to North America rather than the other way around. In this scenario, *Diegoaelurus* and *Machaeroides* might constitute a North American lineage evolving in parallel with Asian machaeroidines. This scenario is contradicted by the phylogenetic topology recovered here. However, as noted above, unambiguous evidence for monophyly of *Diegoaelurus* plus *Apataelurus* is restricted to two highly homoplastic character states associated with hypercarnivory, while support for *Machaeroides* monophyly rests on a single character.

**Implications for Machaeroidine Ecology.**—Machaeroidines are the oldest known carnivorous mammals to show clear adaptations to saber-tooth hypercarnivory. The most

completely known machaeroidine, *Machaeroides eothen*, shows a suite of craniodental features typical of saber-tooth carnivores (*Matthew, 1909*; *Emerson & Radinsky, 1980*; *Van Valkenburgh, 2007*), including the presence of an elongate upper canine, mandibular flange, reduced, hypercarnivorously adapted cheek dentition, reduced coronoid process, ventrally deflected glenoid fossa, and a ventrally expanded mastoid (*Gazin, 1946*). A saber-tooth ecomorphology has evolved iteratively among mammalian carnivores, including in nimravid carnivorans, machairodontine felids, and thylacosmilid metatherians (*Matthew, 1909*; *Emerson & Radinsky, 1980*; *Van Valkenburgh, 2007*). *Barbourofelis* and its relatives represent an additional independent evolution of a saber-tooth ectomorph, either within Nimravidae (*Wang, White & Guan, 2020*; *Barrett, 2021*) or as an independent feliform radiation (*Morales et al., 2001*; *Morlo, Peigné & Nagel, 2004*; *Robles et al., 2013*). Similarly, the North American Miocene felid *Nimravides* may represent an independent acquisition of a saber-tooth morphology within Felidae (*Werdelin et al., 2010*), while extant *Neofelis* may document the early stages of an additional acquisition of a saber-tooth morphology (*Christiansen, 2006*). In taxa with a well-developed saber-tooth morphology, this has been linked to predation on relatively large-bodied prey (*Emerson & Radinsky, 1980*; *Van Valkenburgh, 2007*), although the somewhat divergent specializations of Thylacosmilidae have recently been linked to scavenging (*Janis et al., 2020*). However, *Neofelis* takes relatively small prey, raising the possibility that the prey preferences of taxa showing incipient saber-tooth adaptations may not have differed from contemporary carnivores of comparable size (*Christiansen, 2006*).

Among machaeroidines, the degree of saber-tooth specialization in *M. eothen* is relatively modest. The coronoid process, while reduced and comparable to *Hoplophoneus* (*Bryant, 1991*), remains relatively tall. The lower incisors are relatively small and the lower canine is large and mesiodistally elongate in cross-section (*Gazin, 1946*; *Dawson et al., 1986*). The lower molars retain metaconids and relatively complex talonids and M1 has a substantial protocone, demonstrating incomplete hypercarnivorous adaptation. The mandibular flange of *M. eothen* is also relatively weak, comparable to the conservative nimravid *Dinictis* or the early barbourofeline *Syrtosmilus* (*Scott & Jepsen, 1936*; *Bryant, 1996*; *Werdelin, 2021*). All of these features demonstrate that the saber-tooth adaptation in *M. eothen* was incipient rather than strongly developed.

*Apataelurus kayi* shows evidence of more advanced saber-tooth specialization, with a greatly reduced coronoid process and depressed mandibular condyle comparable to the nimravid *Eusmilus* or even the barbourofeline *Barbourofelis* (*Schultz, Schultz & Martin, 1970*; *Peigné, 2003*; *Barrett, 2016*), consistent with an increased gape necessary to engage an enlarged upper canine. The lower incisors are enlarged and the lower canine reduced, with a mesiodistally compressed cross-section comparable to the incisors, forming an arcade similar to *Hoplophoneus* (*e.g.*, *Scott & Jepsen, 1936*, plate 19). However, the upper canine is unknown in both species of *Apataelurus*, so it is unclear how well-developed it was in this genus. The mandibular flange could provide indirect evidence of canine size, but this feature is also incompletely documented in *Apataelurus*. The dentary of *Diegoaelurus vanvalkenburghae* thus provides the first clear evidence of the degree of development of the canine in an advanced machaeroidine. The mandibular flange is more ventrally expansive

than in either species of *Machaeroides*, approaching the morphology present in early species of the nimravid *Hoplophoneus* such as *H. primaevus* (*Scott & Jepsen, 1936*; *Bryant, 1996*) and consistent with far greater saber-tooth specialization in *D. vanvalkenburghae* than in *Machaeroides*. This, in turn, indicates that *Diegoaelurus* likely shared with other advanced saber-tooth carnivores an ability to hunt comparably large-bodied prey. However, the anterior dentition, with small incisors and a large, mesiodistally broad canine indicates that *D. vanvalkenburghae* lacked the full "canine shear bite" described by *Akersten (1985)*. The elongate base of the flange in *Apataelurus* suggests the potential for even greater saber-tooth specialization in this genus while the sizes of the lower incisors and canine are more consistent with the behavior described by Akersten.

**Machaeroidine Extinction**.—The Ui3 age of *Diegoaelurus vanvalkenburghae* makes it the youngest known representative of Machaeroidinae, along with the taxon represented by CM 2386. By extending the range of machaeroidines to the end of the Uintan, *D. vanvalkenburghae* may link machaeroidine extinction to the substantial faunal turnover event that occurred near the transition between the middle and late Eocene (*Robinson et al., 2004*). During the interval spanning the Uintan-Duchesnean and Duchesnean-Chadronian transitions, many mammalian groups present during the early and middle Eocene disappeared from the North American fossil record, while taxa that would characterize the subsequent White River Chronofauna took their place (*Black & Dawson, 1966*; *Prothero, 1985*; *Robinson et al., 2004*). Documentation of the presence of machaeroidines in Ui3 faunas adds this clade to the former category.

The late Uintan age of *D. vanvalkenburghae* also narrows the temporal gap between the last appearance of Machaeroidinae in North America and the first appearance of another saber-tooth clade, Nimravidae. The oldest definitive North American nimravids are from the middle Chadronian (Ch2) (*Martin, 1998*; *Barrett, 2016*), although there are potential earlier records in the Duchesnean and at the Duchesnean/Chadronian boundary (*Hanson, 1996*; *Poust & Tomiya, 2020*). Accepting the more conservative timing of the First Appearance Datum (FAD) of Nimravidae in North America leaves a temporal gap of approximately five million years between the youngest record of Machaeroidinae and the oldest record of Nimravidae. However, much of the intervening temporal interval is the comparatively poorly sampled Duchesnean. Given the scarcity of machaeroidine remains in much better sampled Bridgerian and Uintan faunas, it is premature to view the lack of a Duchesnean record as definitive evidence of machaeroidine absence. There is at least a potential for competition with early nimravids to have contributed to the extinction of machaeroidines. As with many other topics involving machaeroidine paleobiology and evolution, the limited fossil record of the group precludes confident conclusions.

## CONCLUSIONS

The new taxon described above, *Diegoaelurus vanvalkenburghae*, represents the first machaeroidine named from late Uintan (Ui3) deposits as well as the first North American machaeroidine from outside of Utah and Wyoming. The morphology of the well-preserved, relatively complete dentary of *D. vanvalkenburghae* is noticeably divergent from *Apataelurus*

*kayi*, the only other named North American Uintan machaeroidine. In particular, *D. vanvalkenburghae* has a reduced premolar series, lacking p1 and with a single-rooted p2, contrasting with the unreduced premolar formula of *A. kayi*. On the other hand, *D. vanvalkenburghae* lacks some of the highly derived features of *A. kayi*, including an expanded mandibular flange, greatly enlarged m2, and strongly reduced molar talonids. Combined with the much smaller size of *D. vanvalkenburghae*, the differences from *A. kayi* indicate that at least two machaeroidine lineages were present during the Uintan interval.

Phylogenetic analysis of machaeroidines supports monophyly of middle Eocene taxa, *Diegoaelurus* and *Apataelurus*, with respect to early and earliest middle Eocene *Machaeroides*. The analysis also supports monophyly of *Apataelurus*, linking Chinese *A. pishigouensis* with North American *A. kayi*. The analysis supports monophyly of *Machaeroides*, despite the relatively derived morphology of late early/earliest middle Eocene *M. eothen* compared with early Eocene *M. simpsoni*. Due to the limited material available for all machaeroidines except *Machaeroides eothen*, the topology recovered here is poorly supported. Later taxa are united by highly homoplastic features reflective of hypercarnivory, while monophyly of *Machaeroides* is supported by a single character. More material is needed to test the topology recovered here.

In fact, the present study highlights how poorly documented the machaeroidine fossil record remains. While recent discoveries have improved our understanding of machaeroidine diversity, morphology, and relationships, numerous important questions concerning the evolution, ecology, and extinction of this enigmatic clade remain.

**Institutional abbreviations**

| | |
|---|---|
| **AMNH FM** | Fossil Mammal Collection, American Museum of Natural History, New York City, USA; CM, Carnegie Museum of Natural History, Pittsburgh, USA |
| **FMNH** | Field Museum of Natural History, Chicago, USA; IVPP, Institute of Vertebrate Paleontology and Paleoanthropology, Chinese Academy of Sciences, Beijing, China; |
| **SDSNH** | San Diego Society of Natural History, San Diego Natural History Museum, San Diego, USA |
| **USNM** | United States National Museum of Natural History, Washington, D.C. USA |
| **ZIN** | Zoological Institute, Russian Academy of Sciences, Saint Petersburg, Russia |

## ACKNOWLEDGEMENTS

We are grateful to the staff at the San Diego Natural History Museum involved in the excavation of Jeff's Discovery. Fritz Clarke and others at the SDNHM assisted with preparation. Thanks to Susumu Tomiya, Gabriel Vogeil, Katie McComas, Kesler Randall, and Thomas Deméré for assistance and discussion. Special thanks to Gilbert Boswell, Fidel Rivera, and associates, Department of Radiology, Naval Medical Center San Diego. The research contained herein should not be construed to represent the opinion of the US Navy

or its affiliates. This manuscript has been substantially improved by reviews of an earlier draft by Drs. M. Borths, X. Wang, and L. Werdelin.

### Funding

Ashley Poust was supported by the James R. Colclough Paleontology Endowment, San Diego Natural History Museum. The funders had no role in study design, data collection and analysis, decision to publish, or preparation of the manuscript.

### Grant Disclosures

The following grant information was disclosed by the authors:
Paleontology Endowment, San Diego Natural History Museum.

### Competing Interests

The authors declare there are no competing interests.

### Author Contributions

- Shawn P. Zack and Ashley W. Poust conceived and designed the experiments, performed the experiments, analyzed the data, prepared figures and/or tables, authored or reviewed drafts of the paper, and approved the final draft.
- Hugh Wagner conceived and designed the experiments, analyzed the data, authored or reviewed drafts of the paper, and approved the final draft.

### Data Availability

The list of characters used in the phylogenetic analysis and list of specimen examined are available in the Supplemental File.

The character list, specimen list, and matrix are accessible on MorphoBank: project P4091. Project DOI: 10.7934/P4091, http://dx.doi.org/10.7934/P4091

A three-dimensional model of the specimen described here is available at San Diego Natural History Museum: Project 417128 and Morphosource 000417128.

Link to the SDNHM Project Number: https://3dfiles.sdnhm.org/api/?specimen=38343&name=38343_Dentary_RT&extension=ctm

Link to MorphoSource: https://doi.org/10.17602/M2/M417128.

### New Species Registration

The following information was supplied regarding the registration of a newly described species:

*Diegoaelurus* LSID: urn:lsid:zoobank.org:act:2BE5B4DE-1808-44C4-BF56-89090012ACE6

*Diegoaelurus vanvalkenburghae* LSID: urn:lsid:zoobank.org:act:91F6F357-5E5F-4D5B-BBBD-FE66AB3DF8ED

Publication LSID: urn:lsid:zoobank.org:pub:9946B7AE-5EB5-45BF-953E-B8C245AC806C.

## Supplemental Information

Supplemental information for this article can be found online at http://dx.doi.org/10.7717/peerj.13032#supplemental-information.

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
