# Peer review of "Diegoaelurus, a new machaeroidine (Oxyaenidae) from the Santiago Formation (late Uintan) of southern California and the relationships of Machaeroidinae, the oldest group of sabertooth mammals"

_PeerJ, doi:10.7717/peerj.13032_

## Round 0.1 · original submission · Minor Revisions

Dear authors,

Based on the reviewer comments, I am happy to make a decision of 'minor revisions'.

The comments made by the three reviewers are all constructive, and will strengthen your final manuscript. Please note the comments by reviewer 2 on the phylogenetic analysis. Exhaustive searches should be used for small datasets ('branch and bound' in PAUP terminology).

I look forward to receiving your revised manuscript.

·

Basic reporting

This short paper about a new sabertooth creodont is a pleasure to read. For a group that is so poorly known, even a single lower jaw represents a significant increase of our knowledge. The specimen described in this paper is such an occasion. The paper is a straight forward description and phylogeny, plus some modest discussion about ecomorphology. I find the paper well written, well-illustrated, and the taxon well analyzed and I have little reservations.

Experimental design

I see no problem.

Validity of the findings

A valid new genus and species is named and it is thoroughly documented.

Additional comments

My main suggestion is to add another photo. Since the holotype SDSNH 38343 has a well-developed flange, I suggest that the authors provide an anterior view of the mandible showing the morphology of this structure. As the authors are aware, a flange frequently develops in various sabertooths within Carnivora and often useful as a taxonomic or functional character. Therefore, a more detailed documentation of the flange, in the form of an anterior view, seems warranted, which is often done by sabertooth paleontologists.

I also see a minor problem in the Phylogenetic Results section. The sentence “monophyly of Machaeroides is supported by a single unambiguous synapomorphy, presence of a double-rooted p1 (character 8, state 0)” seems quite implausible to me. Double-rooted p1 is coded as primitive, which I agree, but if so, unless you are trying to demonstrate a character reversal, how could this be a synapomorphy? Merely a shared primitive character, and leaving Machaeroides without a synapomorphy?

Institutional abbreviation: SDSNH suggest adding San Diego Society of Natural History before San Diego Natural History Museum to mitigate the disconnect between SDSNH and the museum.

In general, the ms is well written and quite clean (but I did not have time to read it word for word). It does have an occasional typo, such as line 559, delete “flange” in more ventrally expansive flange.

Xiaoming Wang

·

Basic reporting

The text and figures meet all these criteria, although the text could be better written. Commas are sometimes placed rather haphazardly and the modern 'American' English practice of omitting prepositions (especially 'that') makes it sometimes hard to follow what is referred to in comparative descriptions. An overhaul of the text designed to enhance readability would be welcome.

Experimental design

No issues here and no experiments per se.

Validity of the findings

The paper is an important contribution because machaeoroidine fossils are (almost) as rare as hen's teeth. The results are certainly novel and broadly robust (but see comments below).

Additional comments

I have some specific comments and one more general one that it would be well to address. Specific comments first:

Line 88: I am sure the specimen was measured to the nearest hundredth of a mm, but the accuracy of such measurements using digital calipers is no better than one tenth of a mm (at best). This should be noted to avoid the impression of spurious accuracy.

Line 109: 'more accurate branch lengths' This is completely meaningless. If characters are not parsimony informative they have no impact on a parsimony analysis and should be removed.

Line 124ff: The matrix is small enough that it should be analysed using the exhaustive search function of TnT (it would only take a second or two). That way the shortest tree is guaranteed to be found. In all likelihood it already has been found, but the New Technology algorithms do not guarantee this.

Lines 260, 261, 263: The first to lines include 'paraconid' twice. It is clear from context that the second use must be in error for 'protoconid'. However, on line 263 'paraconid' returns and in this case it is not clear whether 'paraconid' or 'protoconid' is intended.

Line 336: Could the lack of serrations on the lower canine of M. eothen be due to wear? In felid sabertooths serrations tend to be worn away quite rapidly.

Line 369: Could the loss of p1 be caused by the shorter postcanine diastema in D. vanvalkenburghae?

Line 386ff: The synapomorphic state of two-rooted p1 is directly connected to the choice of outgroups. These may or may not be representative of the plesiomorphic state. A change from one to two roots in a lineage where anterior teeth are being reduced seems rather unlikely. Could other outgroups be sought (as the author notes in line 435)? This possibility could be stated earlier.

Now the more general comment: The size and shape of the lower canine is quite unique among sabertooths, particularly forms with a mandibular flange that is as large as than seen in D. vanvalkenburghae. This is the feature that is of most interest in the morphology of the specimen. It's mesiodistal to buccolingual length ratio is very close to that of the clouded leopard (which in turn isn't all that different from any member of Felinae). It contrasts with the condition in Machaeroides and Apataelurus, at least based on how their lower canine alveoli are illustrated in Denison (1938, fig. 5), showing them to be more compressed. This suggests some oddities in the functional apparatus of D. vanvalkenburghae that should be highlighted and discussed.

·

Basic reporting

Overall, the article is an important contribution in the larger effort to understand the evolution of carnivore faunas in North America. I am especially pleased to see the authors have chosen to honor Dr. Blair Van Valkenburg with this particular taxon, which so perfectly captures her field-defining work. The article is well-organized and it appropriately delineates the descriptive, comparative, and phylogenetic results. The table of machaeroidine material is very useful as are the annotations on Figure 2.

Data - In the Materials and Methods section, the authors describe “the 3D imaging” which is not shown in the figures, but is shown as a media image in the MorphoBank project. These models are “available for viewing and download on the NAT Website.” A permanent link (ideally a DOI link) would be useful to include here, or at least a URL for “NAT,” an abbreviation that is not in the Intuitional Abbreviations section. MorphoSource might be a useful resource for accessioning the photogrammetry images and files, and the 3D models that were produced, as MorphoSource provides DOIs for 3D projects.

The MorphoBank project is important, especially the images of the specimens used for the analysis. Given the limited scope of the phylogenetic analysis, I think it would also be useful to include the character-taxon matrix in the Supplemental Data document as well, since it already contains the character list.

Experimental design

Overall, the methods are appropriate for this study. However, it is not clear why the authors generated a novel character-taxon matrix, rather than expanding on the larger matrix published by the lead author, and listed in the References as Zack 2019a. An explanation for this decision in the Materials and Methods section would be appropriate, especially because the authors repeatedly mention a character describing the roots of p1 may be mispolarized, a hypothesis that could be tested by analyzing Machaeroidinae with a substantially larger outgroup sample.

I should also note that I initially assumed the authors had expanded on Zack 2019a when I scanned the manuscript and saw Figure 6, which includes branches for “other Oxyaenidae” and “Hyaenodontidae.” I thought this meant all oxyaenids and hyaenodonts from Zack 2019a were in the analysis. As a lazy researcher, I will often go straight to the temporally calibrated tree figure to get temporal and phylogenetic data simultaneously, but here, that would have mislead me about the breadth of the analysis. Figure 6 should limit its scope to the tree resolved as part of this study, with only the in-group set in temporal context, or it should include only the outgroup OTUs and not their larger clades.

Validity of the findings

The primary finding of this manuscript is that a new genus and species of machaeroidine was discovered in the late Uintan Santiago Formation of California, expanding the presence of the clade beyond the Rocky Mountain Interior. The descriptions and comparisons convince me that this is a new genus and species and that it is appropriately placed in a phylogenetic context, given the selected OTUs, characters, and character scoring. I have a couple of notes (below) on how some of the results are discussed, but this does not detract from the larger conclusions.

Additional comments

Title – Is there a PeerJ stylistic prohibition on putting the genus name and species in the title or the abstract? If not, I think it would be appropriate to include it in both. If the authors would prefer to not put Diegoaelurus vanvalkenburghae in the title, it should at least be included in the abstract. “A new genus” also suggests that this is not a new species, and the specimen was previously described as a new species lumped into a different genus.

Title – The phrase “the relationships of the first sabertooth mammals” does not make it clear that the analysis is limited to machaeroidines. Perhaps, “Diegoaelurus, a new machaeroidine (Oxyaenidae) from the Santiago Formation (middle Eocene) of southern California, and the phylogeny of Machaeroidinae, the oldest group of sabertooth mammals.” Again, because I was familiar with Zack 2019a, I first interpreted the title to mean the analysis would sample nimravids along with machaeroidines.

Line 27 – Insert “taxonomically” between “a” and “small.” At present it reads as if macharoidines are physically small.

Line 28 – Delete “Documented.” As it reads, it sounds like there is an unpublished trove of machaeroidine diversity that the authors are aware of.

Line 34 – Rather than “the only previously named member of the group” perhaps, “the only near-contemporaneous member of the group.” As it reads, the abstract gives the impression Apataelurus kayi is the only named machaeroidine.

Line 72 – Restructure this sentence, so it does not sound like Zack 2019a was written in the late Uintan. Perhaps, “Zack (2019a) described a skeletal association from the Myton Member of the Uinta Formation (late Uintan).” Then start a new sentence with “Wagner (1999) briefly reported…” As structured, the “while” makes it seem like the twenty year gap between the authors was occurring at the same time.

Line 87 – Because Zack (2019b) does not include the detailed dental diagram and description, but instead further refers to Rana et al. 2015 with suggested modification, for clarity and ease of use for future researchers, I suggest changing the reference to Zack (2019b) to “Dental terminology follows Rana et al. (2015) with modifications from Zack (2019b).”

Line 88 – I suggest starting a new paragraph with the details of the photogrammetry process to distinguish these methods from the dental description methods. I also suggest inserting the DOI or URL for the model and photogrammetry materials at the end of the paragraph. Even better would be to deposit the model and images on a site like MorphoSource.

Line 93 – I may have missed it, but what is the “NAT Website”? A URL and defining the abbreviation in the institutional abbreviations would be useful.

Line 98 – See my comments in Experimental Design on adding justification for not using or modifying Zack 2019a in this section.

Line 109 – Branch lengths are not part of a parsimony analysis, and this justification made me think the authors used likelihood or Bayesian methods in addition to parsimony. The phrase “to provide more accurate branch lengths and” should be removed. The justification that the additional characters will help future investigations is sufficient.

Line 113 – It would be helpful to include the character-taxon matrix in the Supplemental Information as well, especially because the matrix is relatively small.

Line 176 – Delete “both”

Line 186 – Perhaps “mental surface” rather than “anterior aspect”? At first I interpreted this as there being no foramina on the distal section of the dentary (including the buccal surface). But maybe there’s no easy way to describe a portion of the jaw that is so rarely illustrated.

Line 224 – Consider replacing “agrees” with “is shared”

Line 225 – Consider inserting a colon after “preserved”

Line 312 – Just a note to again the compliment the authors on clear, vivid description of the material.

Line 313 – In the Reviewer PDF, these section leaders – “Comparisons” and “Phylogenetic Results” – are in the same font and style as the rest of the text. In a scan through the document, they are lost and setting them off more clearly would be useful for the reader. Will PeerJ be formatting these perhaps with a bold type setting or a small section break?

Line 326 – Move the note that known lower molars of M. simpsoni are fragmentary before the statement “m2 does appear to…” When I first read through, I wondered why the authors wouldn’t just measure relative molar size, then was reminded that the record for M. simpsoni is fragmentary. Perhaps, “Known lower molars of M. simpsoni are fragmentary, but the relative molar sizes in the two species do appear to be similar, though m2 does appear to be proportionally larger in D. vanvalkenburghae.”

Line 360 – Delete “North American.” As written is reads as if there is more than one type species for Apataelurus.

Line 380 – Will this results section be more clearly offset in the final publication?

Line 432 – The authors seem to be suggesting it is surprising that they did not recover an anagenetic sequence in the fossil record. If Machaeroides is a genus that has been identified based on shared characters among species, it doesn’t seem surprising that it would be resolved as a clade. The authors repeatedly emphasize the scanty nature of the machaeroidine fossil record. Recovering a simple march through time and taxa would be more extraordinary than the result that was recovered, no? I think this section could be shortened to just report that the results align with stratigraphy and Machaeroides is recovered as a clade.

Line 458 – The concluding sentence of this paragraph summarizes the state of the record succinctly. The proceeding paragraphs could be tightened to the most remarkable observations or important points.

Line 523 – Move the “suite of craniodental features typical of saber-tooth carnivores” forward in this section, from the paragraph beginning at Line 540 to this introductory paragraph. As structured, these key features that are discussed at length later in the section, are buried after a line that appears to narrow its focus to M. eothen.

Line 551 – How are these “advanced saber-tooth specializations” different from specializations to hypercarnivory? The key features for advanced saber-tooth specialization seem to be the height of the coronoid process, the lowered glenoid fossa, and large mandibular flanges. Consider revising or simply deleting the “simplified lower molars” and “enlarged carnassial blade” sections.

Line 560 – This comparative section would benefit from a few direct anatomical comparisons to nimravids and more familiar machairodonine cats. There is the brief mention of Hoplophoneus, but consider more detailed comparisons to support the statement that Apataelurus “show evidence of more advanced saber-tooth specialization.”

Line 608 – Again, “surprisingly” does not seem like the right adverb for describing a clade that was already a genus. It would be more surprising if different Machaeroides species were sister taxa of different Apataelurus species, or if Diegoaulurus was the sister taxon to all other machaeroidines. Consider making the observation that the genus was recovered as a clade, unless there is a body of literature that strenuously makes the argument that Machaeroides is paraphyletic. If so, add that to the introduction and discussion.

Line 630 – The authors thank the staff involved in the excavation of Jeff’s Discovery, but there is no more description of the locality or its history offered in the manuscript. Consider inserting a couple of sentences in the introduction that describe the locality and the efforts made by these staff to document the fauna at SDSNH locality 3276.

Figure 1 – This figure could be used to communicate more detailed information. As it stands, there is a lot of blank space. Consider using color to highlight the early versus middle Eocene sites and to illustrate the rough size and shape of the different basins. There is also room to place the names of the taxa from each basin on the map. Alternatively, is there more detail on the Santiago Formation? A stratigraphic column showing the geological context for the holotype would also be useful to include as part of this graphic.

Figure 6 – See comments above. In essence, I suggest limiting this graphic to in-group (Machaeroidinae). This would also help the reader figure out why only some of the OTUs are connected to the tree.

---

## Round 0.2 · accepted · Accept

Dear authors,

I am glad to say that your manuscript has been accepted for publication in PeerJ. Both reviewers made the 'accept' decision which I agree with.

In due course, the production team will be in contact with you to take you through the proof stages.

Once again, thank you for choosing PeerJ to be your publication venue and I hope you will use us again in the future.

·

Basic reporting

My minor comments on this score have been addressed.

Experimental design

The analysis has been modified to reflect comments that I had in the previous round.

Validity of the findings

No issues here.

Additional comments

After reading the revised text and rebuttal letter I find this manuscript ready to be accepted.

·

Basic reporting

The authors took thoughtful care in addressing the reviewers' comments. Clarifications in the text all flow, and make this manuscript an easy read. The addition of Figure 2 is really helpful, the included links and MorphoSource project, along with the modification of Figures 1 and 7, and the tweak of title all contribute to an even more robust manuscript. I look forward to utilizing this study as we try to sort out the evolution of carnivorous mammals!

Experimental design

Adding the matrix to the supplemental materials and providing the links to specimens are much appreciated.

Validity of the findings

The discussion and conclusions are all consistent with the data. Great work!